# Higher Serum Selenoprotein P Level as a Novel Inductor of Metabolic Complications in Psoriasis

**DOI:** 10.3390/ijms21134594

**Published:** 2020-06-28

**Authors:** Anna Baran, Julia Nowowiejska, Julita Anna Krahel, Tomasz W. Kaminski, Magdalena Maciaszek, Iwona Flisiak

**Affiliations:** 1Department of Dermatology and Venereology, Medical University of Bialystok, Zurawia 14 St, 15-540 Bialystok, Poland; julia.nowowiejska94@gmail.com (J.N.); julita.leonczuk@gmail.com (J.A.K.); iflisiak@umb.edu.pl (I.F.); 2Department of Pharmacodynamics, Medical University of Bialystok, Mickiewicza 2C St, 15-222 Bialystok, Poland; tomasz.kaminski@umb.edu.pl; 3Pittsburgh Heart, Lung and Blood Vascular Medicine Institute, University of Pittsburgh, Pittsburgh, PA 15260, USA; 4Department of Infectious Diseases and Hepatology Medical, University of Bialystok, Zurawia 14 St, 15-540 Bialystok, Poland; mm.maciaszek@wp.pl

**Keywords:** psoriasis, hepatokines, selenoprotein P, cardiometabolic diseases, antipsoriatic treatment

## Abstract

Selenoprotein P (SeP), a member of hepatokines, is involved in the development of various metabolic diseases closely related to psoriasis, but it has not been explored in that dermatosis so far. The study aimed to evaluate the clinical value of serum SeP concentrations in patients with psoriasis and its interplay between disease activity, metabolic or inflammatory parameters and systemic therapy. The study included thirty-three patients with flared plaque-type psoriasis and fifteen healthy volunteers. Blood samples were collected before and after three months of treatment with methotrexate or acitretin. Serum SeP levels were evaluated using the immune–enzymatic method. SeP concentration was significantly higher in patients with psoriasis than in the controls (*p* < 0.05). Further, in patients with severe psoriasis, SeP was significantly increased, compared with the healthy volunteers before treatment, and significantly decreased after (*p* < 0.05, *p* = 0.041, respectively). SeP positively correlated with C-reactive protein and platelets and negatively with red blood counts (*p* = 0.008, *p* = 0.013, *p* = 0.022, respectively). Therapy resulted in a significant decrease in SeP level. Selenoprotein P may be a novel indicator of inflammation and the metabolic complications development in psoriatics, especially with severe form or with concomitant obesity. Classic systemic therapy has a beneficial effect on reducing the risk of comorbidities by inhibiting SeP.

## 1. Introduction

Psoriasis is an autoimmune and inflammatory skin disease affecting 2–4% of the global population, tightly associated with cardiometabolic disorders (CMDs), including obesity, diabetes mellitus (DM), dyslipidemia, non-alcoholic fatty liver disease (NAFLD), metabolic syndrome (MS) and cardiovascular diseases (CVD) [1]. The common background with comorbidities includes not only genetic or immunological aspects but also systemic inflammation, which is crucial in psoriasis pathogenesis [2]. Psoriasis comorbidities constantly lead researchers to look for their newer predictive markers.

The liver affects glucose and lipid metabolism and energy homeostasis by releasing proteins called hepatokines into circulation. Under conditions of high nutrient availability, the liver releases inflammatory markers, including C-reactive protein (CRP) and cytokines into the blood stream. Further research regarding the roles of hepatokines in metabolic diseases could lead to the development of improved targeted strategies for their prevention and treatment [3].

Selenoproteins, belonging to hepatokines, are proteins that include an amino acid selenocysteine in their polypeptide chain and can be found in all lineages of life [4,5]. They are produced in the liver and involved in defense against oxidative stress by taking part in oxidation-reduction reactions neutralizing reactive oxygen species (ROS) [3,4]. It is worth mentioning here that oxidative stress closely related with inflammation participates in the development of pathological processes in psoriasis [2]. Selenoproteins may improve immune response, thanks to their anti-inflammatory, chemo-preventive and antiviral properties [5]. Many disorders seem to be related to selenoprotein metabolism, such as cardiovascular, endocrine, immune and neurological diseases [5]. Furthermore, it was discovered that genetically determined conditions resulting in the reduced production of selenoproteins can cause multiple defects, such as myopathy, male infertility, abnormal levels of thyroid hormones, high sensitivity to UV radiation and loss of neuronal tissue [5,6,7]. The impact of selenium, the most essential part of the selenoprotein molecule, on psoriasis has been previously investigated [8,9,10]. It was observed that there is a decreased selenium level in the whole blood, plasma, red and white blood cells and also the urine of psoriatic patients [8,9,10] and a negative correlation between selenium concentration in RBC (red blood count) or plasma and psoriasis severity, expressed through PASI (Psoriasis Activity and Severity Index) score, was noted [9].

Selenoprotein P (SeP), encoded by SELENOP, located on chromosome 5q31, is an extracellular glycoprotein produced mainly in the liver, involved in transporting and delivering selenium from this organ to body tissues, which contributes to maintaining appropriate selenium levels in tissues [4,11]. Another essential role is taking part in anti-oxidative processes. Furthermore, SeP is considered a multifunctional protein, which appears to have GPx-like (Glutathione peroxidase-like) enzyme activity, peroxynitrite scavenging and metal binding activity [4,11]. Despite its discovery in 1973, it has recently become a particular object of studies on its potential use as a predictor or biomarker of different disease prognoses, pointing to the divergent actions of SeP or even promoting the development of some diseases. The elevated concentration of selenoprotein P was reported in various CMDs, such as obesity, DM, NAFLD and CVD, all closely related to psoriasis [12,13,14]. It has been demonstrated that SeP concentrations are higher in overweight or obese patients [14], especially with visceral obesity [13]. This might point to unexplored links between the selenoprotein and psoriasis, which is multidirectional and linked with obesity, especially through metabolically driven inflammation [2]. Further, psoriasis is considered as an independent risk factor for diabetes, as persons with psoriasis and DM often develop more micro- and macrovascular complications [15]. It is worth emphasizing that selenoprotein P has been proposed as a diabetes-associated hepatokine involved in impaired angiogenesis via the induction of VEGF (vascular endothelial growth factor) resistance in vascular endothelial cells [16,17]. This might point to another link between SeP and psoriasis, while the role of VEGF in that disease has been widely proven [18]. A positive correlation between hepatic SeP expression and type 2 DM in humans has been demonstrated [12]. The hepatokine is involved in insulin signaling downregulation in the liver and muscles and in inhibiting adiponectin synthesis, which, in turn, has been proven to play a significant role in psoriasis [12,19,20]. Increased hepatic SeP has been related to reduced glucose tolerance as well as higher fasting glucose levels [3,21]. On the other hand, as some researchers suggested that increased SeP concentrations may be a secondary condition to DM or reported contradictory outcomes, it seems highly possible that SeP is involved in glucose homeostasis and metabolic disorders, although not many definite conclusions can be drawn [12,22,23].

NAFLD, nowadays reported as the main cause of chronic liver disease in developed countries, affects up to half of the persons with psoriasis [24]. Worse course and higher rate of complications in cases of the co-occurrence of NAFLD and psoriasis have been proven [25]. Elevated SeP levels were noted in patients with NAFLD and visceral obesity [24]. Choi et al. reported on patients in the highest SeP tertile a 7.5 times greater risk of NAFLD than those in the lowest tertile, even after adjustments for age, sex, BMI and other confounding factors, thus suggesting SeP as a potential predictor of the liver disease [13]. Considering the multidirectional relation between NAFLD and psoriasis, it is worth assessing the potency of selenoprotein P in that interplay. SeP also appears to be significantly related to cardiometabolic risk factors, such as waist circumference, VFA (visceral fat area), HOMA-IR (Homeostatic Model Assessment of Insulin Resistance), hs-CRP (high sensitivity C-reactive protein), and baPWV (brachial-ankle pulse wave velocity) values of arterial stiffness [13,24]. The relationship between selenium and dyslipidemia remains uncertain. Although hypercholesterolemia might affect the synthesis of selenocysteine, small amounts of selenium are bound to lipoproteins, and selenium is also reported to modify the activity of lipoprotein lipase in rats, however, it is not fully understood if the same happens in hyperlipidemic patients [26]. However, a few receptors of the LDLR (low density lipoprotein receptor) family have been recognized as selenoprotein P receptors in mice but also in human myoblasts, which might reflect the crosstalk between SeP and lipid metabolism [17,27]. SeP was also investigated as a potential biomarker of the risk of CVD incidence. A strong association between low SeP plasma levels and the risk for all-cause mortality, cardiovascular mortality and a first cardiovascular event in a large group of adults was reported [6]. As it is well known, patients with psoriasis have a shorter survival time compared to the general population, due to cardiovascular disorders [28]. Therefore, it seems advisable to search for pathogenetic relationships and indicators, such as SeP, that would contribute to prolonging life expectancy and preventing comorbidities. To the best of our knowledge there is lack of data concerning the potential role of selenoprotein P in psoriasis pathogenesis. Our aim was to investigate serum SeP levels in patients with flared plaque-type psoriasis and its interactions with disease severity and metabolic or inflammation indicators. Furthermore, we aimed to assess the impact of systemic antipsoriatic therapy on selenoprotein P levels in order to estimate its potency to evaluate the efficacy of methotrexate and acitretin in psoriasis or perhaps contribute to develop newer therapeutic paths.

## 2. Results

The clinical, demographic and laboratory characteristics regarding the study group are summarized in Table 1 and Table 2.

A total of 33 patients with active plaque-type psoriasis, 12 women and 21 men with the mean age of 43.82 ± 16.77 years and 15 age- and sex-matched healthy subjects were enrolled into the study. The mean value of BMI was 27.16 (19.2–42.71) kg/m^2^, as shown in Table 1. The median of basal PASI score was 17.8 (8.5–33.8) points, as shown in Table 2. The median of serum SeP concentration in patients was 7.49 (5.198–49.07) ng/mL before treatment and 7.24 (4.257–34.99) ng/mL after, and was significantly higher compared to the controls: 6.83 (4.34–8.02) ng/mL (*p* < 0.05), as shown in Figure 1a.

After dividing the study group regarding PASI, the protein concentrations were significantly increased in subjects with severe psoriasis (PASI II) before treatment, compared to the controls (*p* < 0.05), as shown in Figure 1b. The comparison of SeP levels between the subgroups, based on the treatment type, shows that patients before treatment are characterized with significantly higher levels of SeP, which were normalized after the treatment, as shown in Figure 1. Therapy with MTX, in contrast to acitretin, resulted in significant changes in SeP levels before and after treatment, compared to controls, as shown in Figure 1c.

SeP level did not correlate with psoriasis severity before the therapy introduction and with total BMI before treatment (*p* > 0.05), as shown in Table 3.

However, SeP levels were associated negatively with PASI without statistical significance, after treatment, as shown in Table 3. With regard to demographic data, SeP did not correlate with the age of the patients (*p* > 0.05), but a downward trend with gender was noted (*p* = 0.085). Of the laboratory indices, strong positive correlations between SeP and CRP and PLT levels (*p* = 0.006 and *p* = 0.046, respectively), and also a negative correlation with RBC (*p* = 0.013) were noted in patients before treatment, as shown in Table 3. All of these associations vanished after treatment; however, after dividing the main group into subgroups treated with ACY or MTX, strong positive correlations between SeP levels and CRP and Total Chol were found (*p* = 0.043 and *p* = 0.046, respectively), as shown in Table 3. No other significant relations between SeP and lipid metabolism indicators, nor glucose or liver enzyme activity were observed. However, some positive tendency with glucose was noted, as shown in Table 3.

The levels of SeP remained unchanged in terms of statistics when comparing the subgroups based on BMI values, however, the BMI III subgroup showed markedly increased levels of SeP before treatment, as shown in Figure 2.

A multivariate linear regression analysis, after considering lipid parameters and blood morphological parameters, revealed that WBC and RBC were independently related to SeP concentration before the treatment (**), as shown in Table 4. (BMI I: n = 8; BMI II: n = 18; BMI III: n = 7).

Interestingly, despite the lack of correlation with the lipid parameters, in patients with elevated LDL or CRP levels, SeP concentrations were significantly higher compared to the controls and patients of normal LDL or CRP levels, as shown in Figure 3.

Regarding selected relations inside the BMI subgroups, we found that in overweight psoriatics (BMI II), before treatment SeP positively correlated with PASI, total platelets number and the levels of hs-CRP, and inversely with RBC, ALT and TG. In obese patients (BMI III), before treatment negative correlations with LDL and RBC were noted, while PLT correlated positively, as shown in Figure 4 on the left panel. After twelve weeks of systemic treatment, previously observed correlations no longer existed, however in the BMI I group, after treatment a strong positive correlation between SeP and RBC, as well as LDL count, and negative associations with PASI score were observed. In the BMI II subgroup, only weak dependency between SeP levels and hs-CRP levels was observed. Interestingly, a negative correlation between SeP levels and RBC count was observed in the BMI III subgroup before treatment, which reversed during the period of treatment, as shown in Figure 4 on the right panel.

## 3. Discussion

In the last decade more and more attention has been paid to hepatokines and their impact on numerous diseases. We were the first to evaluate the potential value of fibroblast growth factor 21 (FGF21) as one of the hepatokines in psoriasis. In a recently published paper, we concluded that FGF21 might be a novel predicting factor of CMD development in patients with psoriasis, especially with severe form or concomitant obesity [29]. These outcomes, together with significant roles in pathophysiological conditions, especially metabolic ones, prompt us to further explore the links with other hepatokines, particularly selenoprotein P. The biological function of SeP is characterized by the selenium transport, but it definitely has wider, unsuspected and still to be elucidated roles.

To the best of our knowledge, there are no studies assessing serum selenoprotein P levels in patients with psoriasis, especially in relation with its systemic therapy. As the first, we attempted to clarify the potential value of this protein in psoriasis. On the one hand, it is a privilege, but on the other a great limitation to conduct full and diligent discussion. In the presented study, the serum selenoprotein P level was significantly increased in patients with psoriasis compared with healthy persons. Further, it was also significantly higher in patients with a severe form of the disease, but also markedly increased in obese patients. It can be assumed that SeP may be a novel potential predictor or perhaps inductor of psoriasis complications, as their co-occurrence significantly rises within the intensity of the disease. Therefore, selenoprotein P might serve as an indicator of inflammation in the general psoriatic population. Further, it may reflect the intensified risk of CMDs, particularly in the most severe and obese patients. Perhaps, in mild-to-moderate psoriasis, the unspecified homeostasis of the protein level is being controlled, or it might not fluctuate as dynamically as in deeper pathological conditions, or it might be some kind of a compensating mechanism.

We must address here the issue of dependence on selenium concentration. We did not evaluate the psoriatics’ selenium intake levels, however its deficiency was reported in those patients [8,9]. Increased SeP concentration in the study group revealed that our study could be partly translated through greater demand for selenium, along with its enhanced utilization for SeP production. Thus, we can assume that selenoprotein P increase might be a secondary effect, rather than a cause, as similarly described in diabetic patients [12,30]. Although the association of selenium deficiency with multiple diseases has been confirmed, many attempts to introduce supplementation resulted in inconsistent conclusions [31]. Moreover, the presence of genetic variations in genes involved in selenium metabolism should be considered, as they affect selenium status in the human body [31]. Further, variations in the *SELENOP* gene results in various responses in the expression of selenoproteins, due to diets containing large amounts of selenium [32]. The role of low selenium level in the diet, resulting in decreased selenoprotein production, has been proven so far in the pathogenesis of different diseases, such as Keshan disease, Kashin–Beck disease, myxedematous endemic cretinism and male infertility [33]. The role of diet in psoriasis is well-established and it is commonly known that persons with psoriasis eat food containing more calories and simple carbohydrates [34]. As mentioned above, they also have decreased selenium plasma concentrations [10]. The research considering the influence of selenium supplementation on psoriatic lesions resulted in divergent clues: some revealed it is not beneficial, but others suggested it may have positive effect on patients’ skin conditions [10,35]. It is also worth highlighting that selenium has its own activity in human organisms, independent from selenoproteins, so, although dietary changes in selenium intake can influence selenoproteins, the relations between these molecules are more complex [36]. Currently, selenium supplementation is not mentioned in the 2019 Joint AAD-NPF guidelines of care for the management of patients with psoriasis, and there are no definitive or general recommendations [37].

Interestingly, the relations between selenium level and CVD risk are often contradictory. Some data point to cardiovascular risk being independent of selenium status, while others to potential preventive selenium supplementation that decreases CVD mortality [6]. Gathering these data, our own results, along with geographical or ethnical differences in selenium status, emphasize the mysteries about selenoprotein P yet to be elucidated.

Notably, serum selenoprotein P levels significantly negatively correlated with red blood cell count. Erythrocytes are important oxygen carriers, taking part in tissue oxygenation homeostasis. Under various pathological and inflammatory conditions, RBC level may decrease inter alia as a consequence of the intrinsic sensitivity of erythrocytes to ROS or disturbed erythropoiesis [38]. A crucial and protective role of selenoproteins in that interplay has been confirmed [38,39]. Given the multidimensional mechanism of these relationships, our results cannot be conclusive. However, additionally noted in multivariate regression analysis, independencies between SeP and RBC and PLT, along with documented links between psoriasis and anemia, further highlight the value of SeP as a modulator of redox homeostasis in psoriasis [40]. Undoubtedly, this requires explored research.

We demonstrated a significant positive correlation between selenoprotein P and CRP or PLT levels. These interrelationships are bi-directional because, in subjects with elevated CRP, the concentration of SeP was significantly higher than in those of normal CRP values. Our outcomes are in line with others who have also noted a positive correlation between SeP and CRP in diabetic subjects [12,41]. Yang et al. additionally found an independent relation of selenoprotein P with carotid intima-media thickness and other CMD indicators, such as as body mass index, waist circumference, systolic blood pressure, triglycerides, glucose, hemoglobin A1c, aspartate aminotransferase and insulin resistance (IR) [41]. Admittedly, we did not show any significant relations between SeP and lipid levels, nor liver enzymes or glucose, but a positive tendency with the latter was seen. However, a strong positive upward trend observed in the hepatokine levels in obese psoriatics points to some links with adiposity. It is consistent with published research, in which higher SeP levels were found in obese persons [14]. Despite the lack of correlations of SeP with lipid levels, interestingly we observed significantly higher levels of the hepatokine in patients with elevated LDL levels compared with those of normal range. These outcomes, together with the abovementioned association of SeP with LDLR, suggest some relationship between the protein and dyslipidemia in psoriasis, which needs to be explored.

As mentioned above, several researchers have reported higher SeP concentration, just like us, in patients with metabolic disorders, such as DM, obesity or NAFLD [13,17]. Mita et al. demonstrated that the excess of plasma selenoprotein P enhances IR but also reduces insulin synthesis through eliminating pancreatic β cells [11]. The pivotal role of insulin resistance in psoriatic march, leading to endothelium dysfunction and CVD, has been widely confirmed [2]. On the other hand, the data are still inconsistent. Although Ko et al. showed significantly increased SeP levels in obese children with NAFLD, it correlated negatively with certain MS components [22]. Similarly, di Giuseppe et al., based on MRI imaging, noted inverse relations of serum selenoprotein P concentrations with several metabolic indices [42]. Furthermore, Altinova et al. did not show any difference in SeP level in diabetic versus non-diabetic pregnant women [23]. Finally, Polyzos et al. in a recently published study revealed decreased SeP levels in patients with definite NASH (nonalcoholic steatohepatitis) compared with controls, and they did not rely on the severity of steatosis, fibrosis or lobular and portal inflammation [43]. Finally, a decreased level of selenoprotein P has been noted in patients with colorectal or prostate cancer and cerebrovascular events [44,45,46]. These contradictory outcomes raise the need for larger studies to clarify the ambiguous effects of SeP. Further, the discrepancies may be due to not yet standardized assay kits used by researchers to quantify SeP levels. Thus, recently, more attention is being paid to a better validation and calibration of the assays used in what might also influence the sexual-dimorphism in selenoproteins [47,48]. In our study group, we found a strong trend reflecting higher SeP concentrations in women, which is consistent with some study outcomes [49,50]. However, the data are striking, also showing no change between sexes in SeP levels or sex-specific and inter-individual differences in selenium or SeP status [48].

To date, there is no research reporting on the influence of antipsoriatic treatment on selenoprotein P. We demonstrated a decrease in serum SeP concentration after total treatment with acitretin or methotrexate and separately with both drugs, which resulted in loss of significance between the controls. Further, therapy with methotrexate caused a more meaningful decrease in SeP level in comparison to acitretin. In patients with severe psoriasis, therapy resulted in a significant decrease in SeP concentration by losing its primary meaningfully elevated level vs. the controls, regardless of the type of the drug. Therefore, we have demonstrated that well-established antipsoriatic treatment suppresses selenoprotein levels in psoriatic persons and particularly in those most affected with psoriasis. Considering the ambiguous effects of SeP and promoting the development of various metabolic diseases obtained decreasing after therapy in general study groups, together with those with severe psoriasis, indirectly indicates its beneficial effect on the inhibition of comorbidities in psoriasis. Further, we might assume that MTX has a more favorable influence than acitretin.

Slightly similar research related to ours reported no superiority of selenium supplementation in patients treated with NB-UVB versus placebo plus phototherapy [51]. In another study, the authors reported no efficacy of supplementation with selenomethionine in patients with psoriasis undergoing topical treatment [8]. Another, still not closely related to ours, was the study of Breedlove et al. who noted that selenium level, as an element of SeP, did not significantly differ from the controls in subjects undergoing therapy with methotrexate [52]. These data cannot be directly compared with the results obtained due to the administration of MTX along with other drugs (CMF chemotherapy: cyclophosphamide, methotrexate, fluorouracil), to patients suffering from breast cancer, and the measurement of selenium plasma, not SeP level [52].

Considering SeP as a potential new predictor or perhaps inducer of increased risk of systemic comorbidities, particularly in severe psoriasis, it might serve as a therapeutic target. Several attempts have been conducted to achieve the suppression of selenoprotein P level, including commonly used drugs. Statins and fibrates, such as PPAR (peroxisome proliferator-activated receptor) alpha agonists, have been reported as decreasing selenoprotein expression [53]. It seems additionally promising, since there are numerous data supporting the beneficial influence of statins and different PPAR agonists on reducing skin lesions in psoriasis, which reflects the role of SeP in that interplay [53,54,55]. There are also reports on the suppressing effect of metformin on selenoprotein P production in the liver, which contributes to the improvement of IR, however this was not investigated on human samples [56,57]. Tajima-Shirasaki N. et al. presented that eicosapentaenoic acid (EPA), a component of ω-3 PUFAs (polyunsaturated fatty acids), inhibits SeP expression in rat hepatocytes [58]. Although, the data concern an animal model, they correlate with the wide literature data proving EPA deficiency in subjects with psoriasis and the beneficial effect of ω-3 PUFA supplementation on their clinical improvement [59,60,61]. Finally, Mita et al. demonstrated that the administration of selenoprotein P-neutralizing monoclonal antibodies to mice with diabetes significantly improved glucose intolerance and IR [11]. Such a novel molecular strategy targeting SeP against the development of DM also suggests a promising approach to treating psoriasis.

We have to point out the limitations of our study, such as, above all, the small sample-sized study group, and even more subgroups divided according to PASI score or BMI. Despite this, the results obtained are encouraging, and in the case of a larger number of patients, they would likely reach more significant values. Therefore, our outcomes should be treated as a prognosis based on preliminary research. To our best knowledge, there is no similar research considering selenoprotein P in psoriatic patients, so we had no data for comparison. Although there are multiple publications regarding the role of SeP in other diseases that could be potentially applied to psoriasis, it is very important to consider the fact that the protein’s activity and biological pathways or sensitivity to selenium deficiency may vary between the tissues and pathological conditions [31,62]. There are still many unknowns about SeP, including its dependence on selenium status or other modifiers in psoriasis. As mentioned above, there is also an urgent need for the validation of assays used for SeP quantification.

## 4. Materials and Methods

In this prospective study, 33 patients (12 women and 21 men) aged 43.84 ± 16.77 years old with a flare of plaque-type psoriasis were recruited at the Department of Dermatology and Venereology, Medical University of Bialystok. The exclusion criteria were other forms of psoriasis or any chronic inflammatory, autoimmune or metabolic diseases, which could affect the results, and a current or five-year history of any neoplasms. None of the patients were under dietary restrictions or received any chronic, systemic or topical treatment for one month prior to enrollment. The Psoriasis Area and Severity Index (PASI) score was determined by the same investigator in all patients who were further divided into two subgroups: PASI I (PASI < 20 points), meaning mild-to-moderate psoriasis, consisting of 23 patients, and PASI II (PASI ≥ 20) including 10 persons with severe form. Body mass index (BMI) was calculated as weight/height^2^ (kg/m^2^). All patients were also subdivided into groups with regard to BMI. BMI I was related to normal weight (BMI 18.5–24.9) and included eight persons. BMI II reflects 18 overweight psoriatics (BMI 25–29.9) and BMI III reflects seven obese psoriatics (BMI > 30). The levels of high sensitive C-reactive protein (hs-CRP), complete blood count (CBC), serum glucose, total cholesterol (Total Chol), HDL (High-density lipoprotein cholesterol), LDL (Low-density lipoprotein cholesterol), triglycerides (TG), and transaminases were evaluated in the study and control group before and after treatment in the study group. The blood samples were collected before initiation and after three months of systemic treatment with methotrexate (MTX) (n = 14 patients), or acitretin (ACY) (n = 19 persons), considering patients’ conditions, tolerabilities and indications, but also considering previous methods of therapy and their efficacy. Serum SeP concentrations were evaluated in relation to normal values collected from 15 healthy age-, sex- and BMI-matched volunteers. The study was approved by the Bioethical Committee of the Medical University in Bialystok (30 November 2017), it was registered with the number: R-I-002/429/2017 and was in accordance with the principle of the Helsinki Declaration. All the participants provided written informed consent before enrollment.

### 4.1. Serum Collection

Blood samples were collected from the study and control groups using Vacutainer tubes, and were left for 30 min to allow clotting before centrifugation for 15 min at 1000× *g*, after which the serum was separated and stored at −80 °C until use. SeP levels were measured using the ELISA kit for Selenoprotein P1, Plasma (SEPP1), Cloud Clone, SEB809Hu. The minimum detectable dose of SeP was less than 0.33 ng/mL, and the standard curve ranges were 0.78–50 ng/mL. Optical density was read at a wavelength of 450 nm. The concentrations were assessed by interpolation from calibration curves, prepared with standard samples provided by the manufacturer.

### 4.2. Statistical Analysis

Normally distributed continuous variables were summarized as mean ± standard deviation (SD), when non-normal distributed data were shown as a median (full range). The distribution of the data was tested using the Shapiro–Wilk normality test. Comparisons between the two groups were made by *t*-test or Mann–Whitney test when the data showed non-Gaussian distribution. One-way ANOVA test was performed before the comparison of individual groups. followed by post-hoc test to evaluate the differences between the analyzed subgroups. Categorical variables were analyzed by the chi-squared test. The relationship between the paired variables was investigated by Spearman’s rank correlation. Multiple linear regression analysis was performed using a stepwise model with a forward elimination procedure to determine the combined influence of variables on particular parameters of the measured system. A two-tailed p-value lower than 0.05 was considered statistically significant. Computations were performed using GraphPad 8 Prism (GraphPad Software; La Jolla, CA, USA).

## 5. Conclusions

Besides the undoubted positive role of selenoprotein P in human organisms, it also has a negative influence on numerous processes and can be somehow be considered as a double-edged sword, probably in psoriasis as well. The presented study, for the first time, demonstrated the potential value of selenoprotein P as an additional marker of inflammation in patients with psoriasis. Higher serum SeP levels may serve as a novel inducing factor of metabolic comorbidities in the general psoriatic population, and especially in most diseased psoriatics and presumably those with concomitant obesity. Increased SeP noted in patients with elevated LDL concentrations points to the role of the hepatokine in the interplay between psoriasis and dyslipidemia, which is yet to be clarified. Furthermore, significant associations between SeP and blood components points to its unknown links between psoriasis and disturbed erythropoiesis. We reported a lowering effect of systemic therapy on selenoprotein P level and especially a significant decrease in SeP in the most severe or obese patients after methotrexate. We can conclude that well-established systemic therapy is sufficiently effective to inhibit the negative effects of SeP on the development of metabolic complications, particularly in severe psoriasis. However, further research on the precise role of selenoprotein P in psoriasis and the development of its comorbidity is needed, as well as an extensive approach to developing new therapeutic options for treating psoriasis, specifically targeting SeP.

## Figures and Tables

**Figure 1 ijms-21-04594-f001:**
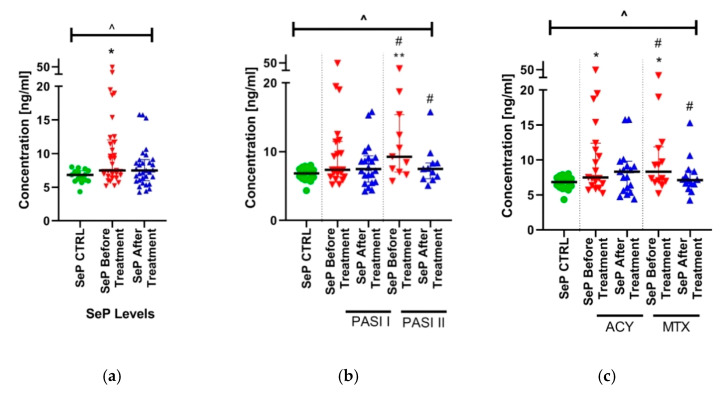
The levels of SeP in the study group before and after total treatment (**a**) compared to controls and divided into subgroups based on PASI (**b**) and undergoing therapy separately with acitretin and methotrexate (**c**). */**/ means the existence of statistically significant difference between patient single group compared to controls with *p* < 0.05; *p* < 0.01, respectively. ^ means the existence of a statistically significant difference between all the groups calculated using ANOVA with *p* < 0.05. # shows the statistical significance between controls and marked patient subgroups when compared using ANOVA with *p* < 0.05. SeP—selenoprotein P. (PASI I: n = 22; PASI II: n = 11; ACY: n = 19; MTX: n = 14).

**Figure 2 ijms-21-04594-f002:**
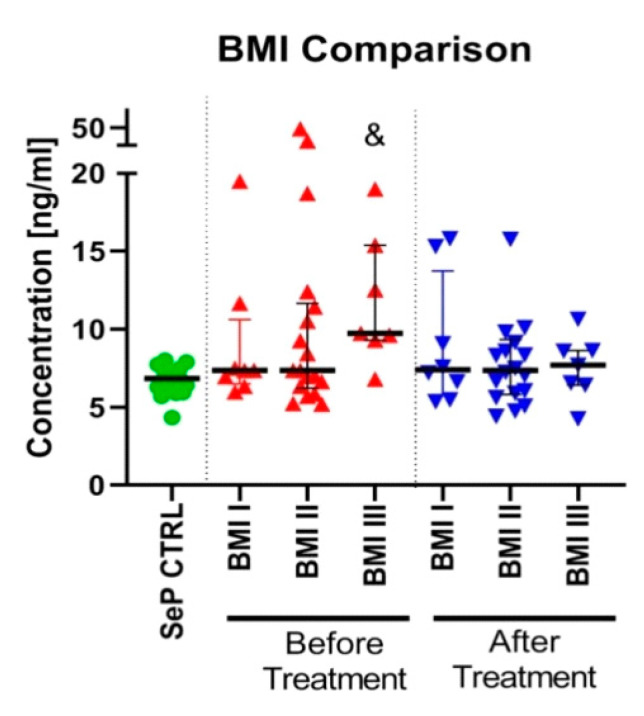
The median values of SeP serum levels in BMI subgroups before and after treatment. & indicates the existence of a trend due to a small number of “n” in the subgroups (*p* < 0.1).

**Figure 3 ijms-21-04594-f003:**
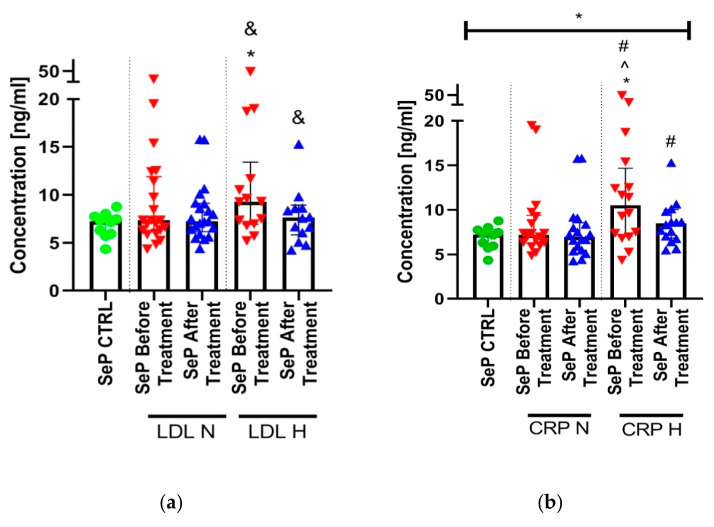
The levels of SeP in LDL normal and elevated subgroups (**a**) or in hs-CRP normal and elevated subgroups (**b**) before and after treatment. (LDL N: n = 19; LDL H: n = 4; CRP N: n = 7; CRP H: n = 16). * means *p* lower than 0.05 when comparing any single subgroup to the controls. ^ shows *p* < 0.05 when compared both subgroups before treatment. & and # mean the existence of statistical significance at level lower than 0.05 when analyzing SeP CTRL and both subgroups in LDL H/CRP H using ANOVA. The asterisk above the graph (b) shows the *p* < 0.05 when comparing all included subgroups using ANOVA.

**Figure 4 ijms-21-04594-f004:**
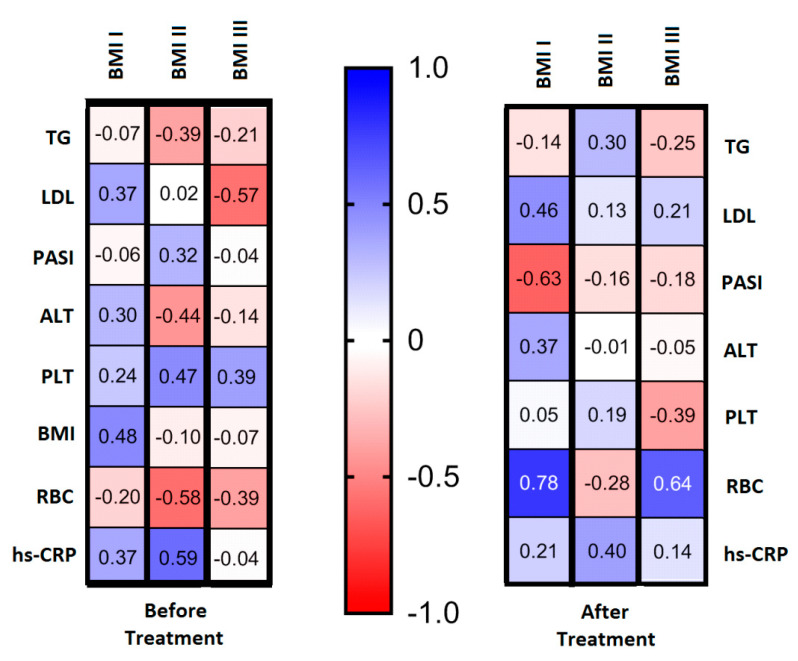
Chosen correlations inside BMI subgroups: before and after total treatment with use of Spearman’s rank correlation.

**Table 1 ijms-21-04594-t001:** Basal characteristic of control group and patient group.

Parameter	Controls (n = 15)	Patients (n = 33)
Sex (M/F)	8/7	21/12 NS
Age (years)	33 ± 15.25	43.84 ± 16.77 NS
Height (cm)	172.1 ± 8.4	173.8 ± 9.15 NS
Weight (kg)	73.3 ± 15.3	81.1 ± 12.47 NS
BMI ratio	23.1 (19.4–31.2)	27.16 (19.2–42.71) NS

NS, non-significant; M/F, male/female ratio; BMI, body mass index.

**Table 2 ijms-21-04594-t002:** Basal characteristic of the patient group before and after treatment in total and separately with both drugs. (Before treatment: n = 33, after treatment: n = 33, ACY: n = 19, MTX: n = 14).

Characteristics	Before Treatment	After Treatment	After Acitretin	After Methotrexate
WBC (×10^3^/ml)	7.56 ± 2.18 ^	7.02 ± 1.78 ^	7.54 ± 2.06 ^	6.31 ± 1.02 **^
RBC (×10^3^/ml)	4.61 ± 0.52	4.62 ± 0.51	4.62 ± 0.48	4.74 ± 0.57
PLT (×10^3^/ml)	233.2 ± 65.78	235.7 ± 58.03	238.1 ± 56.11	233.5 ± 61.87
Chol [mg/dl]	170.5 ± 28.53	181.1 ± 32.14	182.05 ± 25.94	182.1 ± 41.31
HDL [mg/dl]	47.55 ± 12.05	47.32 ± 19.79	52.21 ± 22.25	42.25 ± 14.65
LDL [mg/dl]	104.2 ± 23.47	107.2 ± 26.37	104.8 ± 26.64	111 ± 26.21
TG [mg/dl]	122.2 ± 50.87 ^	150.5 ± 73.63*^	132.8 ± 66.82 ^	174.28 ± 79.9 *^&
Glucose [mg/dl]	84 (68–227)	87 (71–243)	86 (72–243)	88 (71–115)
ALT [U/L]	17 (8–76)	17 (8–111)	16 (9–111)	18 (8–89)
AST [U/L]	19 (12–71)	17 (10–112)	21 (13–112)	18 (10–75)
CRP [mg/L]	2.98 (1–57.68)	1.87 (1–15.8) *	1.87 (1–15.8) *	2.09 (1–5.3) &
PASI before and after treatment	17.8 (8.5–33.8) ^^^	10.3 (6-23.5) ***^^^	10.4 (6–23.5) **^^^	10.15 (6.1–14.4) ***^^^

*/**/*** means the existence of statistically significant difference between values after and before treatment with *p* < 0.05; *p* < 0.01; *p* < 0.001, respectively. ^/^^^ means the existence of statistically significant difference between all the 4 groups calculated using ANOVA with *p* < 0.05; *p* < 0.001, respectively. & means the existence of trend due to low *n* per subgroup. PASI, psoriasis area and severity index; RBC, red blood cells; PLT, platelets; WBC, white blood cells; TG, triglycerides; HDL, high-density lipoproteins; LDL, low-density lipoproteins; CRP, C-reactive protein; ALT, alanine transaminase; AST, asparagine transaminase.

**Table 3 ijms-21-04594-t003:** Correlations between baseline parameters and SeP in sera of the study group before and after total treatment and both drugs separately.

Parameter	Before Treatment R, (*p* Value)	After Treatment R, (*p* Value)	After Acitretin R, (*p* Value)	After Methotrexate R, (*p* Value)
PASI	0.085/NS	*−0.314 /NS (0.075)*	−0.353/NS	−0.238/NS
BMI	0.208/NS	-	-	-
CRP	0.472/(0.006) **	*0.308/NS (0.08)*	0.451/(0.43)	0.217/NS
WBC	0.137/NS	0.178/NS	0.168/NS	−0.257/NS
RBC	−0.428/(0.013) *	0.129/NS	−0.110/NS	0.426/NS
PLT	0.379/(0.046) *	0.027/NS	0.322/NS	−0.328/NS
Total Chol	−0.093/NS	0.291/NS	0.435/(0.046) *	0.116/NS
HDL	−0.167/NS	−0.216/NS	−0.307/NS	−0.138/NS
LDL	−0.007/NS	0.175/NS	0.335/NS	0.026/NS
TG	−0.205/NS	0.091/NS	0.150/NS	0.033/NS
Glucose	*0.307/NS (0.082)*	0.005/NS	−0.278/NS	*0.458/(0.083)*
ALT	−0.081/NS	0.052/NS	0.018/NS	0.002/NS
AST	0.059/NS	0.070/NS	−0.013/NS	0.082/NS

*/** indicates statistical significance with *p* values <0.05; <0.01. *Italic* font indicates a trend (*p* < 0.1).

**Table 4 ijms-21-04594-t004:** Parameters that independently predict the levels of SeP before treatment.

Parameter	|t| Value	*p* Value	*p* Value Summary
Total Chol	0.9201	0.266	NS
TG	1.449	0.1597	NS
HDL	0.3704	0.7142	NS
LDL	0.1206	0.9050	NS
WBC	3.693	0.0011	**
PLT	0.7761	0.4450	NS
RBC	4.136	0.0003	***

Included Variables: SeP, lipid metabolism parameters, morphological parameters, glucose. Goodness of Fit: Multiple R = 0.7488; R squared = 0.5608; Adjusted R squared = 0.4388.

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
