# Peer review of "Higher Serum Selenoprotein P Level as a Novel Inductor of Metabolic Complications in Psoriasis"

_ijms, 2020, doi:10.3390/ijms21134594_

Round 1

Reviewer 1 Report

The article fits the aims of the journal and falls within the journal purpose. The manuscript is a study upon the variation of selenoproteinP (SeP) with treatment in psoriasis patients.

The introductory section is well written and thoroughly documented, presenting SeP in a broader picture deliniated by metabolic conditions such as diabetes, non-fatty liver disease, cardiovascular ailments, presenting in an unbiased manner the stat-of-the-art level of knowledge, pinpointing both clear and less clear aspects of the role SeP plays within the pathogenesis of the aforementioned disease spectrum, also identifying knowledge gaps that could be exploited in research. One of these gaps is the role of SeP in psoriasis, which authors aim to explore.

In this respect, the authors aim to investigate serum SeP levels in patients with flare-ups of plaque psoriasis, to assess whether and to what extent the SeP levels vary along disease severity, among other metabolic and inflammation parameters, and also to evaluate how SeP is altered by systemic treatment in these patients.

However, minor changesshould be taken into consideration:

Abstract: authors are advised to define abbreviations at first appearance in text; in abstract and also elsewhere

Number of patients treated withacitretin (n1..) and MTX respectivelly (n2=……) are not evident from neither Table 2 nor elsewhere. This should be corrected, as the reader would benefit from knowing how manypatients were on the same treatment arm.

Also, the choice/rationale by which patients were asigned to either MTX or acitretin has been evasively mentioned; it would be important to detail upon this methodological step, so the reader to be capacitated to judge whether the asignment criteria could have had an influence over entire study.

Also, the rationale behind inclusion and exlcusion criteria must be detalied some more (i.e. reasons to chose/exclude patients).

Methodological weaknesses: controls were a decade youngerthan patients (rendering the pair matching rather modest, even if age intervals overall are intersecting); an explanation for the choice to pair rather unsimmilar groups would be of interest

Please rephrase “before and aftertotaltreatment and after both drugs 
separately”, in order to make clearer.

  • Figure 2. The median values of SeP serum levels in BMI-subgroups before and after treatment. & - means the existence of trend due to a small number of “n” in the subgroup – Figure 2 is not inserted in the manuscript, at supposed place

Statistics-wise, the authors analyse a wide number of parameters on same patients, and moreover splitted the patients in subgroups (which in this case also happened to be rather small, leading to necessity of exact tests; we cannot know that, partly because we cannot know from the manuscript text how many patients were in each group, e.g. high LDL/normal LDL, high vs normal CRP…); authors might add a short explanation how they mitigated the risk of  multiple comparisons in these particular conditions, which could usually lead to falsely significant associations (falsely detecting an effect).

A comprehensive table depicting mean/median values of analysed parameters (in order to be helpful/significant also to the clinicianeyeballing the values)–and numbers of patients within categories and subcategories (not just correlations and p-values) would be a great added value to the article, so authors would be advised to insert such a table or at least mention where appears in text the number of patients in each group/subgroup.

A replacement/rephrase for the word “psoriatic/ psoriatics” could be considered, as would be of benefit to the article.

Also, English spelling must be carefully checked throughout the entire manuscript.

The article is well written, the authors named in a rational manner the limitations of the study. Authors also showed directions for further research in the field.

Overall,  I consider it a very useful contribution to the journal. Therefore, I recommend the manuscript for being published after suggested minor changes have been taken into consideration by the authors.

Author Response

Thank you very much for your valuable time and reviewing the manuscript. It has been supplemented with detailed suggestions. All changes are matched with red color.

  1. The abbreviations has been defined at first appearance in the text.
  2. Inclusion and exclusion criteria have been a bit more explained. In order not to extend the text and to answer the reasons for inclusions/exclusions: we enrolled patients with flare-ups of psoriasis. These patients required implementation of systemic treatment as we intended to evaluate the interplay with selenoprotein P. Further, these were active stages of psoriasis with most indicative data to obtain.
  3. Number of patients treated with particular drug: Thank you for the remark, we’ve missed these important data which have been put in the text.
  4. Answering to the choice/rationale by which patients were assigned to either MTX or acitretin: As it was stated in the manuscript when choosing the particular drug we considered patients’ condition, tolerability and indications but also previous methods of therapy, including also its efficacy.

Controls were in fact younger than patients, however the difference was statistically insignificant. We correlated both values ​​with SeP in both groups (controls and patients) and the results were as such: patients: Sep vs Age: R = 0.151; P = 0.042 | SeP vs Weight: R = 0.007; P = 0.9676 Control: SeP vs Age R = 0.102; P = 0.62 | SeP vs. Weight R = 0.08; P = 0.88. Thus, it seems that SeP does not depend on the above parameters. In addition, young people are more likely to participate in such studies and what is important are also simply healthier - thanks to this the control group was a fully healthy group.

  1. Figure 2 has been replaced. Thank you.
  2. Numbers of patients in each subgroup have been inserted in the text.
  3. “..authors might add a short explanation how they mitigated the risk of  multiple comparisons in these particular conditions, which could usually lead to falsely significant associations (falsely detecting an effect).”

Thank you for this comment. Taking into account comparatively small amount of subjects in the studied subgroups our results are subjected to the possibility of misinterpretation of the data. To avoid the occurrence of false-positive results in the presented study we made two different types of analysis - ANOVA to check the significance between studied subgroups, then to evaluate the difference between two given subgroups Student t-test or Mann-Whitney test has been used (depends on normal or skewed distribution) P value lower than 0.05 is considered as statistically significant. Moreover, after statistical analysis, all the results were subjected to checking the statistical power of performed calculations. Presented data showed at least 75% power to detect a difference between means of high scientific significance level (alpha) of 0.05 (two-tailed). Thank you again for bringing this issue to our attention. 

  1. A replacement/rephrase for the word “psoriatic/ psoriatics” has been considered and placed in the text.

Once again, thank you for your contribution to our research.

Kindest regards.

Reviewer 2 Report

This is a very interesting article, a small study, including an extensive review of data available to date on SeP.  Results are clear, however and statistically significant. To establish SeP as a biological marker much larger studies are needed and the authors have made this clear.

 Remarks aiming at improving the manuscript:

  1. Figure 2 is missing, the legend is available
  2. Supplementary materials (line 405) have not been provided
  3. Line 174: There is no “ ^/^^^” in Tables 1, 2 or 3
  4. Patients’ mean weight is higher than controls’. How is it established that it is psoriasis and not the differences in body weight that is related to the differences observed in SeP levels?
  5. In Figure 3, there are 3 patients (2 with normal CRP and LDL and 1 with high CRP and LDL) that exhibit an only partial reduction of CRP and LDL. Are these the same 3 patients in both graphs? Could this be an indication of partial resistance to treatment? Is there any additional information concerning these 3 patients?
  6. Text is clear, however minor grammar gaps exist that need improvement:
  • Line 29: “Most diseased” makes no sense
  • Line 247: “In recenty published paper”: read “In a recently published paper
  • Line 254: “the more” makes no sense
  • Lines 260-261: “within the skin involvement” makes no sense
  • Lines 271-272: “Although confirmed association of selenium deficiency and multiple diseases” makes no sense
  • Line 341: “the is no research” better reads: “there is no research”
  • Line 358: “with own results” makes no sense

Author Response

Thank you very much for your valuable time and reviewing the manuscript. It has been supplemented with detailed suggestions. All changes are matched with red color and they’ve been put in the manuscript.

  1. Figure 2 is missing, the legend is available – the figure has been inserted, thank you.
  2. Supplementary materials (line 405) have not been provided- all the data was included in the main body of the paper.
  3. The legends of  the tables have been corrected.
  4. The difference in mean weight and age between patients and controls is not statistically significant therefore this should not affect the results obtained. Considering meaningful involvement of SeP in the development of various metabolic diseases closely related with psoriasis along with the significant increase in serum SeP level obtained in our study highlight its important role in psoriasis as well. We correlated both values ​​with SeP in both groups (controls and patients) and the results were as such: patients: Sep vs Age: R = 0.151; P = 0.042 | SeP vs Weight: R = 0.007; P = 0.9676 Control: SeP vs Age R = 0.102; P = 0.62 | SeP vs. Weight R = 0.08; P = 0.88. Thus, it seems that SeP does not depend on the above parameters. In addition, young people are more likely to participate in such studies and what is important are also simply healthier - thanks to this the control group was a fully healthy group.
  5. In Figure 3, there are 3 patients (2 with normal CRP and LDL and 1 with high CRP and LDL) that exhibit an only partial reduction of CRP and LDL. Are these the same 3 patients in both graphs? Could this be an indication of partial resistance to treatment? Is there any additional information concerning these 3 patients?

Yes, on both graphs the patients that are characterized with excessive high levels of SeP protein are the same. It might be the evidence of partial resistance to the treatment or might be an indicator of underlying conditions that were not diagnosed. This issue has been included in the description of the results.

    6. The minor grammar gaps have been corrected according to the      suggestions. Thank you.

Once again, thank you for your contribution to our research.

Kindest regards. 

Round 2

Reviewer 1 Report

All suggestions from previous version have been taken care of in an adequate manner, the manuscript has been revised properly, therefore I recommend the article for being published in the journal, representing an useful contribution.

Reviewer 2 Report

x